# Human premotor areas parse sequences into their spatial and temporal features

**Katja Kornysheva[1,2]\*, Jörn Diedrichsen[1]**

[1]Institute of Cognitive Neuroscience, University College London, London, United Kingdom; [2]Department of Neuroscience, Erasmus Medical Centre, Rotterdam, Netherlands

**Abstract** Skilled performance is characterized by precise and flexible control of movement sequences in space and time. Recent theories suggest that integrated spatio-temporal trajectories are generated by intrinsic dynamics of motor and premotor networks. This contrasts with behavioural advantages that emerge when a trained spatial or temporal feature of sequences is transferred to a new spatio-temporal combination arguing for independent neural representations of these sequence features. We used a new fMRI pattern classification approach to identify brain regions with independent vs integrated representations. A distinct regional dissociation within motor areas was revealed: whereas only the contralateral primary motor cortex exhibited unique patterns for each spatio-temporal sequence combination, bilateral premotor areas represented spatial and temporal features independently of each other. These findings advocate a unique function of higher motor areas for flexible recombination and efficient encoding of complex motor behaviours.

\*For correspondence:
k.kornysheva@ucl.ac.uk

**Competing interests:** The authors declare that no competing interests exist.

**Reviewing editor**: Jody C Culham, University of Western Ontario, Canada

## Introduction

Skilled performance in music, speech, or sports often involves long sequences of movements, which demand a precise sequential activation of different muscles in time. The ordering of these muscle activations—and hence the ordering of the movements of different body parts in space—is here referred to as the 'spatial feature' of a sequence. Additionally, movement sequences are often characterized by a stereotypical temporal structure or rhythm—their 'temporal feature'. The latter can either emerge spontaneously as part of chunk formation (*Sakai et al., 2003*), or be directly relevant to the goal of the sequence, as in musical performance, dance, or speech (*Shin and Ivry, 2002*; *Lewis and Miall, 2003*; *Repp, 2005*; *Kotz and Schwartze, 2010*; *Bläsing et al., 2012*; *Grahn, 2012*; *Penhune and Steele, 2012*). One of the hallmarks of human motor performance is the ease with which experts can modify the temporal and spatial features of learned motor skills. For example, a pianist is able to play the same tune using different variations of the rhythm, and a fluent speaker can change separately the word order or the rhythmic profile of speech for effective communication. How is such flexibility in skilled actions achieved neurally?

There has been a long-standing debate on whether a dedicated representation of temporal structure of skilled movements exists, or whether it is tightly integrated with a representation of its spatial features (*Conditt and Mussa-Ivaldi, 1999*; *Shin and Ivry, 2002*; *Ullén and Bengtsson, 2003*; *Medina et al., 2005*; *Spencer and Ivry, 2009*; *Ali et al., 2013*). Recent work suggests that spatio-temporal trajectories of movements can be learned and produced by a dynamical network of neurons that encodes patterned muscle dynamics, instead of by representing different parameters of a movement sequence separately (*Laje and Buonomano, 2013*; *Shenoy et al., 2013*). This neural implementation has been advocated for the primary motor and premotor cortices and implies that temporal features are stored inseparably from the specific movement trajectory trained. From this perspective, a spatial

**eLife digest** Once a pianist has learned to play a song, he or she can nearly effortlessly reproduce the sequence of finger movements needed to play the song with a particular rhythm. A skilled pianist can also improvise, pairing the same keystrokes with a different rhythm or playing the same rhythm with a slightly different sequence of keys. This ability to flexibly modify and recombine sequences of physical movements in space and time enables humans to exhibit great creativity in music, language, and many other tasks that require motor skills. However, the underlying brain mechanisms that allow this flexibility are only beginning to be explored.

Some scientists have theorized that networks of brain cells in the parts of the brain that control movement store a sequence in time and space as one inseparable unit. However, this theory doesn't explain why pianists and other skilled individuals can separate and recombine the physical movements and timing of a sequence in new ways. An alternate idea is that the brain captures the information necessary to execute a series of physical movements separately from the timing at which the movements are to be carried out. This would allow these features to be put together in new ways.

Kornysheva and Diedrichsen taught a group of volunteers a series of finger movements paired with particular rhythms. Half the volunteers performed the task using their left hand and the other half with their right hand. After training the volunteers performed better when producing sequences they had been trained on, even in trials where either the rhythm or the finger sequence was slightly changed.

The volunteers were also asked to perform the trained movements while their brain activity was monitored in a functional magnetic resonance imaging (fMRI) machine. Kornysheva and Diedrichsen looked for areas that showed similar patterns of increases and decreases in activity whenever a particular sequence was performed. This identified areas that showed unique patterns for each trained sequence combination of finger movements and rhythm, which could be distinguished from areas where the activity patterns for sequences remained similar across rhythms or across finger movements.

Kornysheva and Diedrichsen found that a region of the brain that controls movement encodes sequences on the opposite side of the brain from the moving hand. In this part of the brain, the movement and timing were encoded together as one unit. However, in premotor areas—which are known to help individuals to plan movements—the timing and the finger movements appeared to be encoded separately in overlapping patches on both sides of the brain. This automatic separation appears to be a fundamental function of the premotor cortex, enabling behavioural flexibility and the storage of complex sequences of movements in space and time.

sequence performed with two different temporal profiles would constitute two distinct behaviours and demand the training of independent neural generators.

Alternatively, the motor system may parse movement sequences into their constituent spatial and temporal features, which then are represented independently. Such an encoding scheme would explain the ability of both animals and humans to flexibly recombine learned temporal patterns with a new spatial sequence and vice versa (**Ullén and Bengtsson, 2003**; **Ali et al., 2013**; **Kornysheva et al., 2013**).

Neurally, sequence representations are characterised by the occurrence of sequence-specific tuning. For example, neurons in the supplementary motor area (SMA) vary their firing rate for specific movement transitions and whole sequences of movements rather than for individual movements in a sequence (**Tanji and Shima, 1994**). Inactivating the SMA, the primary motor cortex, the putamen or the dentate nucleus in monkeys disrupts sequential behaviour whilst sparing individual actions (**Tanji and Shima, 1994**; **Shima and Tanji, 1998**; **Hikosaka et al., 1999**; **Lu and Ashe, 2005**), with a similar effect demonstrated for the SMA/pre-SMA in humans with non-invasive stimulation techniques (**Gerloff et al., 1997**; **Kennerley et al., 2004**). Recent evidence in primates also argues for the existence of neurons tuned to specific temporal intervals between movements in the same area, with a subset also tuned to the position of an interval in a sequence (**Merchant et al., 2013**). However, it remains unknown whether these neurons are simply part of a dynamical network that represents

spatial and temporal features in an integrated manner, or whether independent populations of neurons encode spatial and temporal features in isolation.

Here, we used fMRI to study the sequential tuning of individual voxels in the human brain. We hypothesized that following training, specific neuronal sub-populations will become differentially active for different sequences, as has been observed in neurophysiological studies for spatial sequence features (*Tanji and Shima, 1994*). If such sequence-specific tuning is sufficiently clustered, it should be detectable with the relatively low spatial resolution of fMRI (*Kamitani and Tong, 2005*; *Swisher et al., 2010*). Using a classification approach to evaluate these subtle differences in the local patterns of brain activity during sequence production, a recent imaging study (*Wiestler and Diedrichsen, 2013*) indeed showed that such sequential tuning can be detected in the human brain in a range of motor and pre-motor areas. This study, however, did not reveal whether and how these areas represented spatial or temporal features of sequences.

To this end, we developed a visually paced motor learning paradigm. Participants were trained on nine sequences consisting of unique combinations of three spatial and three temporal features (*Figure 1*). Half the participants were trained on the right and half on the left hand to probe whether possible differences between hemispheres reflected hemispheric specialisation or the difference between contra vs ipsilateral encoding. First, by looking at behavioural generalisation, we show transfer of trained temporal and spatial features to new combinations. Second, by employing separate classification procedures of fMRI voxel activity patterns and testing for generalization of patterns across temporal or spatial contexts, we were able to dissociate independent spatial and temporal from integrated representation profiles across the human motor system.

## Results

### Learning and transfer of sequence features

We used a visually cued motor learning task to induce and assess the acquisition of sequences involving finger movements (*Kornysheva et al., 2013*). Subjects were trained to produce nine finger sequences that were unique combinations of three temporal and three spatial sequence features (*Figure 1*) randomly generated for each subject. Half the participants trained and performed the sequences with the left, the other half with the right hand. Over the course of 3 days of training the force responses on the keyboard triggered by the visual stimulus became faster (*Figure 2A*). The average reaction time (RT) decreased from 410 ms (SD: 70) to 288 ms (SD: 48, *Figure 2B*). To assess the specificity of the improvement in motor performance to the trained sequences, we also

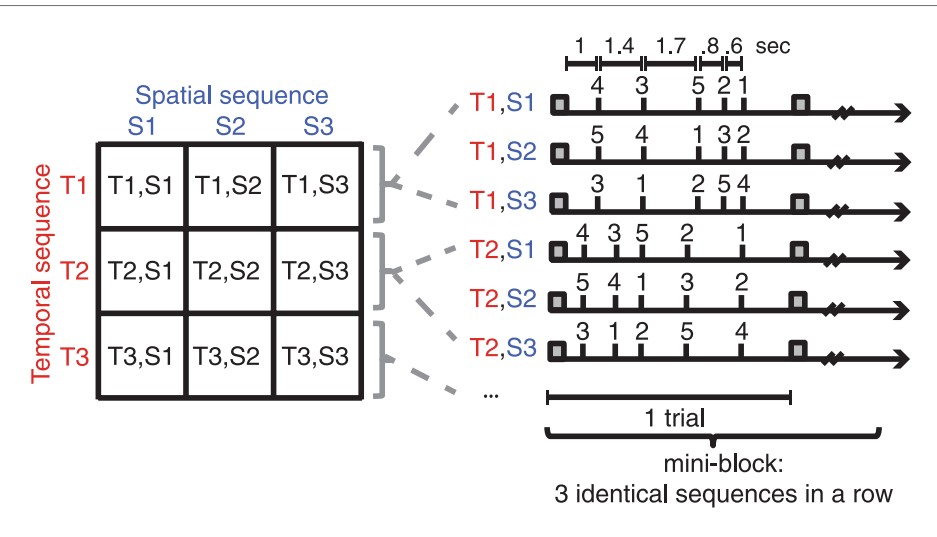

**Figure 1**. Subjects were trained on nine sequences, which were unique combinations of three spatial (finger order) and three temporal sequence features. Sequences were presented in mini-blocks of three trials in a row. Each sequence began with the presentation of a warning cue (square).

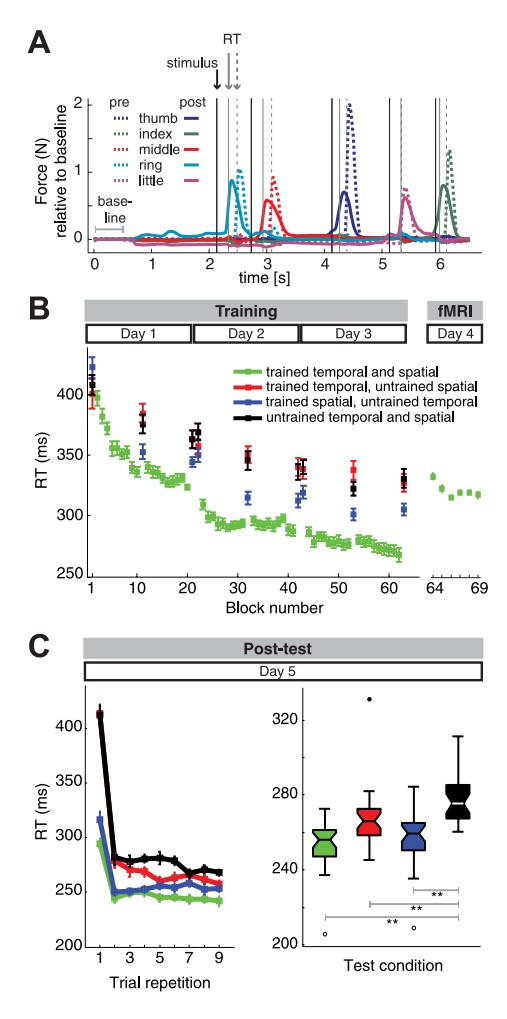

**Figure 2**. Reaction time (RT) results. (**A**) Two trial examples of force traces show faster finger responses to visual stimuli after ('post') as opposed to the beginning of training ('pre'). (**B**) Subjects showed general and sequence-specific learning during the training of the combined temporal and spatial sequences. The RT remained relatively stable across the fMRI session runs, albeit overall higher than at the end of training outside the MRI environment. (**C**) Post-test results. Left panel: repeating sequences nine times in the test phase yielded an immediate RT decrease for trained spatial sequences (blue) relative to untrained sequences (black), and only delayed RT differences for trained temporal sequences (red), in line with previous results (***Kornysheva et al., 2013***). Right panel: a boxplot displaying RT results across subjects and all sequence repetitions in the post-test revealed significant RT advantages for the trained sequence, as well as the trained spatial and trained temporal feature conditions when compared to untrained sequences, suggesting that both the finger order and their relative timing were represented independently. A double asterisk (**) indicates a significant difference between conditions
*Figure 2. Continued on next page*

tested participants on sequences composed of untrained spatial and temporal features. Subjects also reduced the RT for untrained sequences from 456 ms (SD: 90) to 346 ms (SD: 64), which suggests a general effect of visuomotor learning. However, the reduction was significantly smaller than that for the trained sequences ($F_{(1,30)}$ = 13.342, p=0.001), and there was no interaction with the group (right-hand-trained vs left hand, $F_{(1,30)}$ = 1.235, p=0.275). Overall error rates across conditions paralleled the RT findings (***Figure 2— figure supplement 1***). For trained sequences the error rate reduced from 46.1% (SD: 26.7) to 6.3% (SD: 4.9). For untrained from 52.1% (SD: 31.6) to 29.5% (SD: 21), suggesting that the RT findings were not due to a change in the speed-accuracy trade-off. During the fMRI session we only tested the nine trained sequences. The RTs increased as compared to the end of training (***Figure 2B***, *t*(31) = 6.57, p<0.0001). This occurred despite the subjects' familiarity with a supine position on a mock MRI bed during training and may be related to the fMRI noise during the performance of the task, the lack of any auditory movement feedback occurring, as well as a more restricted mobility in an MRI environment. Importantly, however, across subjects individual RTs during fMRI session strongly correlated with RTs for trained sequences in the last training block (*r* = 0.717, p<0.0001), indicating that the responses reflected consistent measures of behaviour.

On the day following fMRI, we conducted a post-test to assess whether participants would be able to utilize both the learned spatial and temporal features when these were paired with novel untrained features. Based on previous studies (***Shin and Ivry, 2002***; ***O'Reilly et al., 2008***; ***Brown et al., 2013***; ***Kornysheva et al., 2013***), we expected evidence only for spatial, but not for temporal feature transfer in the first three trials. Indeed, during the training phase, in which each sequence was repeated only three times in a row (***Figure 2B***), and during the first trials in the post-test (***Figure 2C***) the temporal transfer condition was not performed faster than untrained control sequence. However, consistent with two previous experiments (***Kornysheva et al., 2013***), a delayed RT advantage for the temporal transfer condition emerged after a few repetitions of the new sequence combination (***Figure 2C***, left panel). Averaged over all nine repetitions in the post-test, sequences which combined a trained temporal ($F_{(1,28)}$ = 12.963, p=0.001) or spatial ($F_{(1,28)}$ = 20.830, p=0.0001) feature with an untrained feature were performed faster than a repetition of a completely novel sequence (***Figure 2C***, right panel).

*Figure 2. Continued*

with p<0.01. Errorbars in the lineplots and maximum and minimum RTs (boxplot whiskers) are corrected for differences in the mean RT across individuals, and therefore represent the interindividual variability of relative RT differences across conditions. In the boxplot, upper and lower edges signify the 75th (third quartile) and 25th percentile (first quartile), respectively. The median is designated as a horizontal white or black line in the box. Outliers (equal or above 3*interquartile range above the third quartile or below the first quartile) are depicted as filled circles, suspected outliers (1.5*interquartile range above the third quartile and below the first quartile) are depicted as unfilled circles, respectively.

The following figure supplement is available for figure 2:

**Figure supplement 1**. Error rate paralleled the RT results during the training (**A**) and fMRI showing clear sequence-specific advantages for the trained sequences, as well as sequences which retained the spatial features.

This suggests that training of timed finger sequences automatically results in independent storage of its spatial and temporal features. ANOVAs also revealed that these effects did not differ between the left and right-hand-trained groups (p>0.127). These results replicate earlier findings and can be explained by a model in which the temporal representation acts as a multiplicative go signal on a concrete spatial plan (*Kornysheva et al., 2013*); while both temporal and spatial sequences are represented independently, the temporal representations act at the output stage as a modulator on a spatial expectation. Thus, without a spatial expectation, temporal knowledge does not have an effect. For this reason, temporal transfer cannot manifest itself immediately, but only after a plan for the next movement has been formed (*Hikosaka et al., 1999*; *Penhune and Steele, 2012*).

## Average activation increases

Compared to rest, the visually cued production of trained finger sequences yielded increased activity in motor and visual areas. To analyse the imaging data from the left-hand and right-hand groups together, we grouped the hemispheres according to whether they were ipsi- or contralateral to the moving hand. As would be expected from earlier studies of visually trained skilled sequence production (*Wiestler and Diedrichsen, 2013*), increased activation was observed in the contralateral primary motor cortex (hand knob area of M1) extending into the dorsal premotor cortex (PMd), the ipsilateral lobule V-VI of the cerebellum, bilateral (pre) supplementary motor area (SMA/pre-SMA), bilateral ventral premotor cortex (PMv), left dorsolateral prefrontal cortex (dlPFC), and visual areas—left posterior precuneus and secondary visual cortex (V2), as well as right associative visual and occipito-temporal cortices.

## Classification approach to determine independent and integrated representations

To determine the tuning characteristic of population of voxels in different cortical areas, we utilized a system of four different cross-validated classification analyses ('Materials and methods'). The overall classifier (*Figure 3A*) was trained and tested on all nine sequences (albeit from different imaging runs), and distinguished between all sequences independently of their component features. As confirmed by simulations of voxel activity patterns (*Figure 3E*, black bar), this classifier shows above-chance accuracy whenever there are any reliable differences between the nine activity patterns, independently of the underlying tuning functions, and therefore can detect any sequence representation.

In contrast, a region that contains an independent representation of the order of finger presses in space should show consistent activity patterns for spatial features of sequences, independent of their temporal features. To detect such patterns, a spatial classifier was trained on trials where the three spatial features were combined with two different temporal features, and then tested on data in which these spatial features were combined with a new temporal feature (*Figure 3B*). Therefore, this classifier can only yield above-chance accuracy, when the voxels show consistent tuning for the spatial features, independently of the temporal structure of finger sequences. The temporal classifier (*Figure 3C*) was defined in the same way, but classified temporal features combined over two spatial features, and tested on the left out spatial feature. Finally, the integrated classifier (*Figure 3D*) tested for a nonlinear interaction between the tuning for temporal and spatial features by subtracting out information that could be explained by each component (spatial or temporal) separately. The classifier therefore only detects regions that show unique, idiosyncratic patterns for each of the nine sequences. Our pattern simulations demonstrated that this set of four classifiers could sensitively reveal the type of the underlying representation. Importantly, we could distinguish between a region containing a unique,

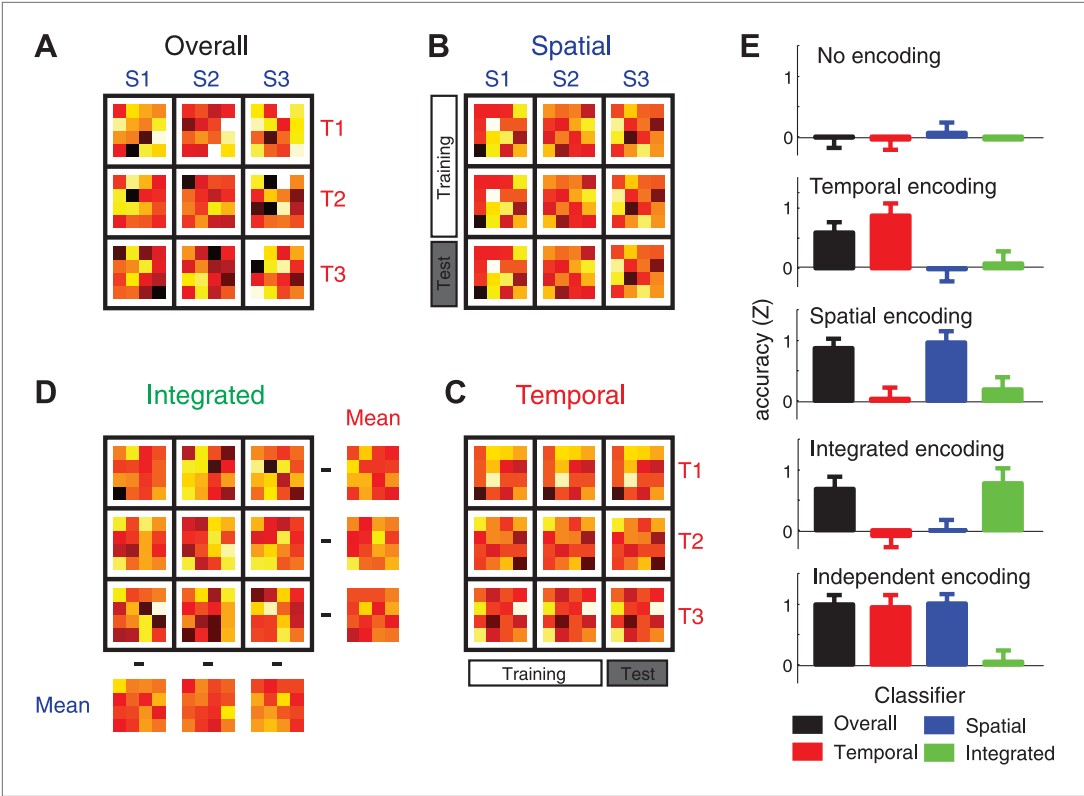

**Figure 3**. Four classification procedures were employed to classify the voxel pattern of each searchlight (160 voxels, here reduced to 16 units for illustration purposes). (**A**) To test whether a voxel searchlight contained any sequential information, the overall classifier distinguished between the nine sequences independently. Classification was always cross-validated across imaging runs ('Materials and methods'). (**B**) To determine encoding of the spatial feature, the classifier was trained on data involving only two of the three temporal sequences, and tested on trials from a left-out imaging run in which the spatial sequences were paired with the remaining temporal sequence. (**C**) The temporal classifier followed the same training-test principle, but in an orthogonal direction. (**D**) The integrated classifier detected nonlinear encoding of the unique combinations of temporal and spatial features that could not be accounted for by linear superposition of independent encoding. The spatial and temporal mean patterns for each run were subtracted from each combination, respectively, to yield a residual pattern, which was then submitted to a nine-class classification. (**E**) Classification accuracy of the four classifiers on simulated patterns (z-transformed, chance level = 0). Results indicate that the underlying representation can be sensitively detected by contrasting the overall, temporal, spatial, and integrated classifiers. Importantly these classification procedures can differentiate between a non-linear integrated encoding of the two parameters as opposed to the overlap of independent temporal and spatial encoding.

integrated, representation of the two sequential features (in which case only the integrated classifier should be above chance, *Figure 3E*, 'Integrated encoding') and a region containing a superposition of independent temporal and spatial sequence representations (in which case both independent classifiers within one region are above chance, *Figure 3E*, 'Independent encoding').

## Overall sequence representations

To determine which cortical regions showed any differences between the activity patterns for the nine unique combinations of temporal and spatial features, we utilized the overall classifier. We found significant above-chance classification accuracy in the hand knob area of the contralateral M1 extending into primary sensory area (S1), PMd and PMv, the contralateral anterior insula, the ipsilateral Lobules V–VI of the cerebellum (*Figure 4—figure supplement 1*), the bilateral SMA, superior parietal and extrastriate cortices, the ipsilateral medial M1, and inferior parietal cortex (*Figure 4A*; *Table 1*). Overall, the above threshold clusters were more widespread on the contralateral than on the ipsilateral side. These results were in line with the regions found in an earlier study (*Wiestler and Diedrichsen, 2013*),

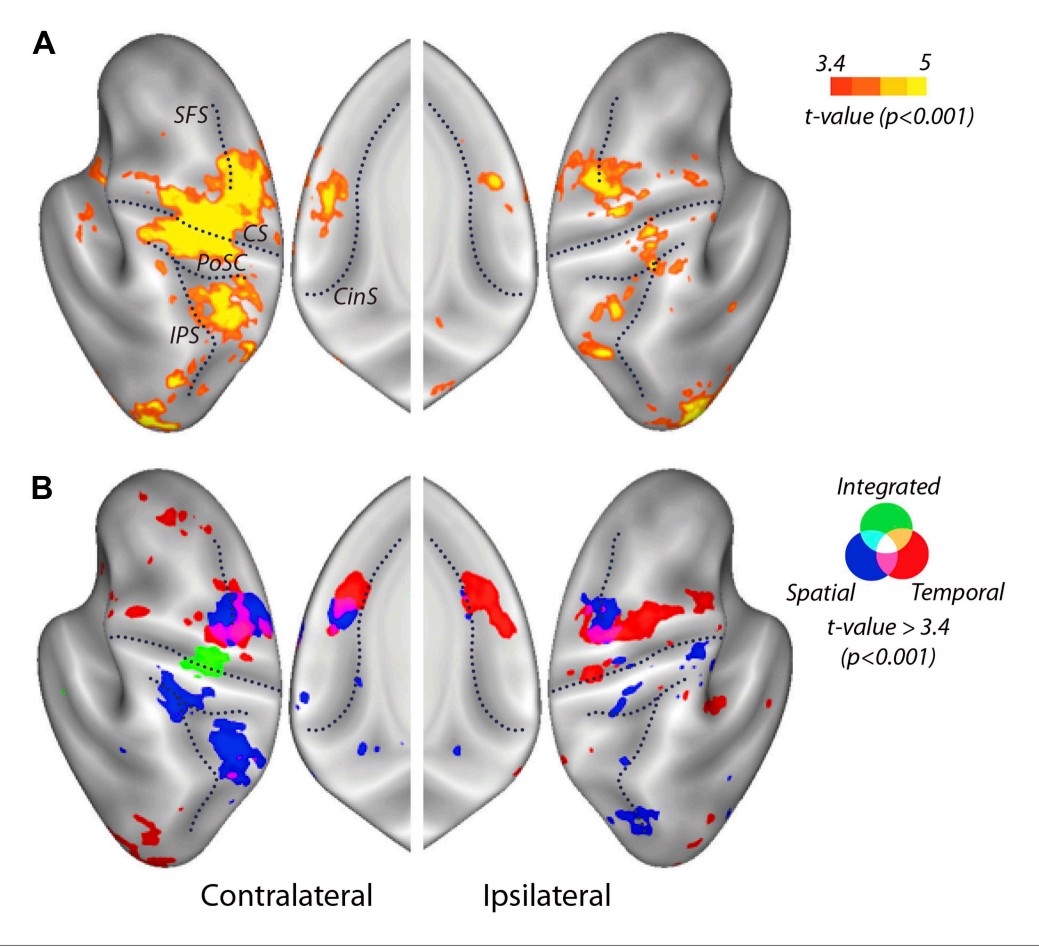

**Figure 4**. Searchlight classification results shown on an inflated representation of the cortical surface. (**A**) Group t-values indicate regions in which the overall classifier performed significantly above chance. (**B**) Significant group-level above chance classification of spatial (blue), temporal (red), and integrated (green) classifiers. Results are presented at an uncorrected threshold of $t(31) > 3.37$. $p<0.001$. CinS, cingulate sulcus; CS, central sulcus; IPS, intraparietal sulcus; PoSC, postcentral sulcus; SFS, superior frontal sulcus.

The following figure supplements are available for figure 4:

**Figure supplement 1**. Searchlight classification results in the cerebellum.

**Figure supplement 2**. Mean searchlight classification accuracy results displayed as in *Figure 4*, split by group trained on the right and left hand.

**Figure supplement 3**. Classification accuracy of the main response function and temporal derivative.

**Figure supplement 4**. Maximum force for finger 1(thumb) to 5 (pinkie) during fMRI.

which used faster finger sequences that were produced from memory. Thus, our results indicate that similar encoding can be found for visually paced sequences involving longer temporal intervals between finger presses. From the overall classifier, however, we cannot yet determine how different features of the sequences were encoded.

## Integrated and independent encoding of sequence features

We then determined which regions encoded the nine sequences with a unique activity pattern without any consistent patterns for temporal or spatial features/components alone, using the integrated

**Table 1.** Areas showing above-chance classification accuracy for the decoding of sequences and their spatial and temporal features

| Classifier | Area (Brodmann area) | Area (cm²) | P_cluster | Peak $t_{(31)}$ | MNI | | |
|---|---|---|---|---|---|---|---|
| | | | | | X | Y | Z |
| Overall | Contralateral | | | | | | |
| | M1/PMd/PMv (BA4/BA6) | 43.25 | <0.001 | 9.40 | −36 | −22 | 53 |
| | Superior parietal (BA40/BA7) | 15.80 | <0.001 | 5.82 | −32 | −54 | 56 |
| | Extrastriate vis cortex (BA18) | 8.53 | <0.001 | 5.75 | −27 | −90 | −1 |
| | Extrastriate vis cortex (BA19) | 2.35 | 0.002 | 5.57 | −36 | −83 | 27 |
| | SMA (BA6) | 3.86 | <0.001 | 5.27 | −8 | −12 | 57 |
| | | 2.22 | 0.002 | 4.73 | −42 | −82 | −13 |
| | Anterior insula (BA48) | 1.35 | 0.036 | 4.35 | −35 | −10 | −2 |
| | | 1.81 | 0.008 | 4.32 | −42 | 2 | 12 |
| | Occipitotemporal area (BA37) | 1.75 | 0.01 | 4.10 | −40 | −62 | −11 |
| | Ipsilateral | | | | | | |
| | Extrastriate vis cortex (BA19) | 20.84 | <0.001 | 5.77 | 34 | −89 | −6 |
| | PMd (BA6) | 12.72 | <0.001 | 5.24 | 21 | −12 | 60 |
| | Superior parietal (BA5) | 3.76 | <0.001 | 5.19 | 19 | −55 | 61 |
| | Superior parietal (BA7) | 3.27 | <0.001 | 4.92 | 30 | −59 | 46 |
| | Medial M1 (BA4) | 4.84 | <0.001 | 4.86 | 14 | −40 | 59 |
| | Occipitotemporal area (BA37) | 1.83 | 0.014 | 4.55 | 45 | −70 | 6 |
| | Extrastriate vis cortex (BA19) | 2.55 | 0.002 | 4.22 | 32 | −75 | −11 |
| | SMA/Pre-SMA (BA6/BA32) | 1.39 | 0.05 | 3.93 | 8 | 18 | 49 |
| Integrated | Contralateral | | | | | | |
| | M1 (handknob, BA4) | 5.89 | <0.001 | 5.39 | −33 | −23 | 59 |
| Spatial | Contralateral | | | | | | |
| | Superior parietal (BA7) | 10.00 | <0.001 | 6.93 | −31 | −56 | 60 |
| | PMd (BA6) | 9.66 | <0.001 | 6.20 | −31 | −13 | 53 |
| | Inferior parietal (BA40) | 6.00 | <0.001 | 5.78 | −39 | −36 | 37 |
| | SMA (BA6) | 2.69 | 0.002 | 5.70 | −9 | 1 | 54 |
| | Ipsilateral | | | | | | |
| | PMd (BA6) | 5.23 | <0.001 | 4.68 | 29 | −2 | 47 |
| | Inferior parietal/occipital (BA39/BA19) | 3.98 | <0.001 | 4.31 | 33 | −66 | 34 |
| Temporal | Contralateral | | | | | | |
| | SMA (BA6) | 3.47 | <0.001 | 5.74 | −8 | 9 | 48 |
| | PMd (rostral BA6) | 6.38 | <0.001 | 5.53 | −24 | −15 | 58 |
| | Extrastriate vis cortex (BA18) | 11.00 | <0.001 | 4.58 | −29 | −92 | −5 |
| | Extrastriate vis cortex (BA19) | 2.35 | 0.006 | 4.34 | −35 | −82 | 10 |
| | Ipsilateral | | | | | | |
| | PMd (rostral, BA6) | 9.78 | <0.001 | 5.98 | 23 | −9 | 49 |
| | PMv (BA6) | 5.19 | <0.001 | 5.28 | 51 | −6 | 24 |
| | Posterior cingulate (BA23) | 2.44 | 0.006 | 4.83 | 9 | −30 | 31 |
| | Pre-SMA/anterior cingulate (BA32) | 2.73 | 0.004 | 4.79 | 9 | 34 | 42 |

*Table 1. Continued on next page*

*Table 1. Continued*

| Classifier | Area (Brodmann area) | Area (cm²) | P$_{cluster}$ | Peak t$_{(31)}$ | MNI X | Y | Z |
|---|---|---|---|---|---|---|---|
| | PMd (caudal BA6) | 1.75 | 0.034 | 4.66 | 20 | −26 | 57 |
| | Extrastriate vis cortex (BA19) | 1.75 | 0.034 | 4.13 | 42 | −85 | 1 |

Results of surface-based random effects analysis (N = 32) with an uncorrected threshold of t(31) > 3.37, p<0.001. p (cluster.) is the cluster-wise p-value for the cluster of that size. The p-value is corrected over the cortical surface using the area of the cluster (**Worsley et al., 1996**). The cluster coordinates reflect the location of the cluster peak in MNI space.

classifier. The only region that carried such integrated, non-additive encoding was the cortical output area—the M1 (handknob area) contralateral to the hand involved in producing the sequences (**Figure 4B**, green, **Table 1**). This region was clearly visible in the contralateral motor cortex of both the right and the left-hand groups (**Figure 4—figure supplement 2**). To test the encoding in motor-related areas in more detail, we analysed the data in four symmetrically defined regions of interest (ROIs): primary motor cortex (M1), dorsal (PMd), ventral premotor cortex (PMv), and supplementary motor area (SMA). Mean integrated encoding reached only significant above chance level in the contralateral M1 (**Figure 5**, p=0.0005, Bonferroni correction at p=0.05/8=0.0063), but not in any of the premotor regions (p>0.155).

We then searched for regions in which voxels would show consistent tuning for temporal or spatial features of the sequence, independently of the respective other component. For the spatial and temporal classifiers we found highest encoding outside M1, particularly in premotor, as well as in parietal areas. The spatial classifier detected consistent patterns related to the order of the finger presses that remained unchanged when executed with different temporal features. Spatial classification accuracy was significantly above chance in the contralateral SMA, bilateral PMd, as well as superior and inferior parietal lobes (**Table 1**). The temporal classifier detected representations of temporal sequences, which did not change across different spatial sequences (orthogonal to the spatial classification analysis). Clusters in bilateral PMd, contralateral SMA, ipsilateral PMv, anterior and posterior cingulate, and bilateral extrastriate visual areas were significant after correction for multiple tests (**Table 1**).

These results suggest that while contralateral M1 exhibits mostly integrated encoding of temporal and spatial sequence features, the two sequence components are represented independently in the bilateral premotor cortex. An ROI (M1 vs premotor) × hemisphere (contralateral vs ipsilateral) × classifier (integrated vs independent spatial and temporal) × group (right vs left hand) mixed ANOVA indeed revealed a significant ROI × hemisphere × classifier interaction ($F_{(1,30)}$ = 12.808, p=0.001). This effect did not interact with group (p=0.75), suggesting that although the effect was less pronounced for left-hand-trained participants, the distribution of integrated and independent sequence encoding was similar for left and right hand sequence production (**Figure 5—figure supplement 1**).

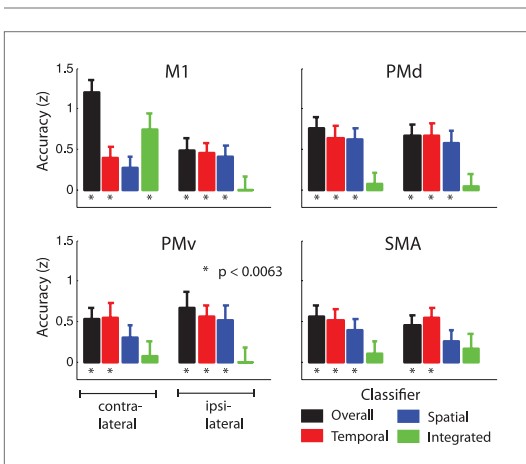

**Figure 5**. Classification accuracy (*z-values*) in anatomically and symmetrically defined motor regions of interest (ROI). Integrated classification accuracy was significant above chance level in contralateral M1 only, whereas temporal and spatial classifiers showed higher accuracy in premotor areas, in a partly overlapping manner.

The following figure supplement is available for figure 5:

**Figure supplement 1**. Classification accuracy as in **Figure 5** split by group trained on the right and left hand.

The difference in representation can be better appreciated in *Figure 6A*, which shows the level of temporal, spatial, and integrated sequence feature encoding on a cross-section running from rostral PMd to the caudal end of the occipito-parietal junction (cf. *Figure 6—figure supplement 1* for profiles split by right and left hand groups). In contrast to temporal and spatial encoding, integrated encoding peaked at the level of the central sulcus in the contralateral hemisphere. To test for differences in the distribution of integrated and independent encoding, we used a Center of Gravity (CoG) analysis. We determined the CoG for integrated and independent (averaged over spatial and temporal) classification accuracies on the precentral part of the cross-section. Indeed, we found a more caudal CoG for integrated, and a more rostral CoG for independent encoding in the contralateral hemispheres of both left- and right-hand groups (*Figure 6C*). A hemisphere (contralateral vs ipsilateral) × classifier (integrated vs independent) × group (right vs left hand) mixed ANOVA revealed a hemisphere × classifier interaction ($F_{(1,30)}$ = 6.417, p=0.017), but no interaction with group (p=0.409). For the postcentral part of the cross-section, we found the reverse pattern—a more caudal CoG for independent as compared to integrated encoding (*Figure 6D*). Again the hemisphere (contralateral vs ipsilateral) × classifier (integrated vs independent) interaction was highly significant ($F_{(1,30)}$ = 13.394, p=0.001), and did not interact with group (p=0.088). These results therefore clearly suggest a difference in how primary motor and premotor, as well as parietal areas represent spatio-temporal finger sequences.

Although temporal and spatial features were represented independently in premotor and parietal cortex (i.e., these regions showed specific activity patterns for one feature, independent of the respective other feature), spatially these representations overlapped to a certain degree, especially in caudal PMd (*Figure 4B*). It is only through the use of multivariate analysis techniques that we were able to distinguish such overlap of independent representations from the integrated representation in the primary motor cortex. Inspection of the temporal and spatial representation maps, however, also suggests a difference in where temporal and spatial features are localised. Especially in the ipsilateral premotor cortex, it appears that temporal encoding was more pronounced in the ventral, whereas spatial representations are more evident in the dorsal premotor cortex. This gradient can be seen more clearly on a profile plot of encoding in the premotor cortex (*Figure 6B*). A CoG across lateral premotor cortex indeed revealed that temporal encoding was centred more ventrally and spatial encoding more dorsally (*Figure 6E*). A hemisphere (contralateral vs ipsilateral) × classifier (temporal vs spatial) × group (right hand trained vs left hand) mixed ANOVA showed a main difference between classifier ($F_{(1,30)}$ = 5.836, p=0.022), but no interaction with group (p=0.687) or hemisphere (p=0.678).

In summary, we found that outside of primary motor cortex, both temporal and spatial features of a learned movement sequence are represented independently, albeit in partly overlapping areas. Furthermore, we found a functional gradient with temporal representation being stronger in ventral and spatial representation stronger in dorsal premotor areas.

## Voxel patterns reflect sequence-related encoding rather than finger-related encoding

One potential concern regarding multivariate analysis for fMRI data is that the voxel patterns used for classification may simply reflect differences in the temporal profile of activation rather than differences in the spatial activation patterns related to sequence-specific encoding. Since voxels in M1 are known to show differential tuning for isolated finger movements (*Diedrichsen et al., 2013b*), some voxels may show higher activity early in each sequence, while others would peak late in the sequence.

However, there is good evidence to suggest that this effect cannot account for the results reported here. First, if the classification reflected tuning to individual finger movements, all classifiers should show the highest accuracy in the contralateral hand area of S1 and M1, where finger representations are the strongest (*Wiestler et al., 2011*). This, however, could not be observed, as evidenced by a differential distribution of temporal, spatial, and integrated encoding across the brain. Second, each sequence trial was produced three times in a row (mini-block), so that the main response function was elevated across the execution of three sequences and did not return to baseline before the end of the last trial (*Figure 4—figure supplement 3A*). It therefore should be insensitive to differences in the temporal activation profile within each execution. In contrast, the temporal derivative of the main response function included in our first level general linear model (GLM), captured variations of the temporal profile within each trial. For example, the derivative would indicate whether a voxel was activated more in the early or late phase of each sequence. For the main classification analysis, we discounted the derivative, as we wanted to isolate differences in spatial activity patterns, rather than

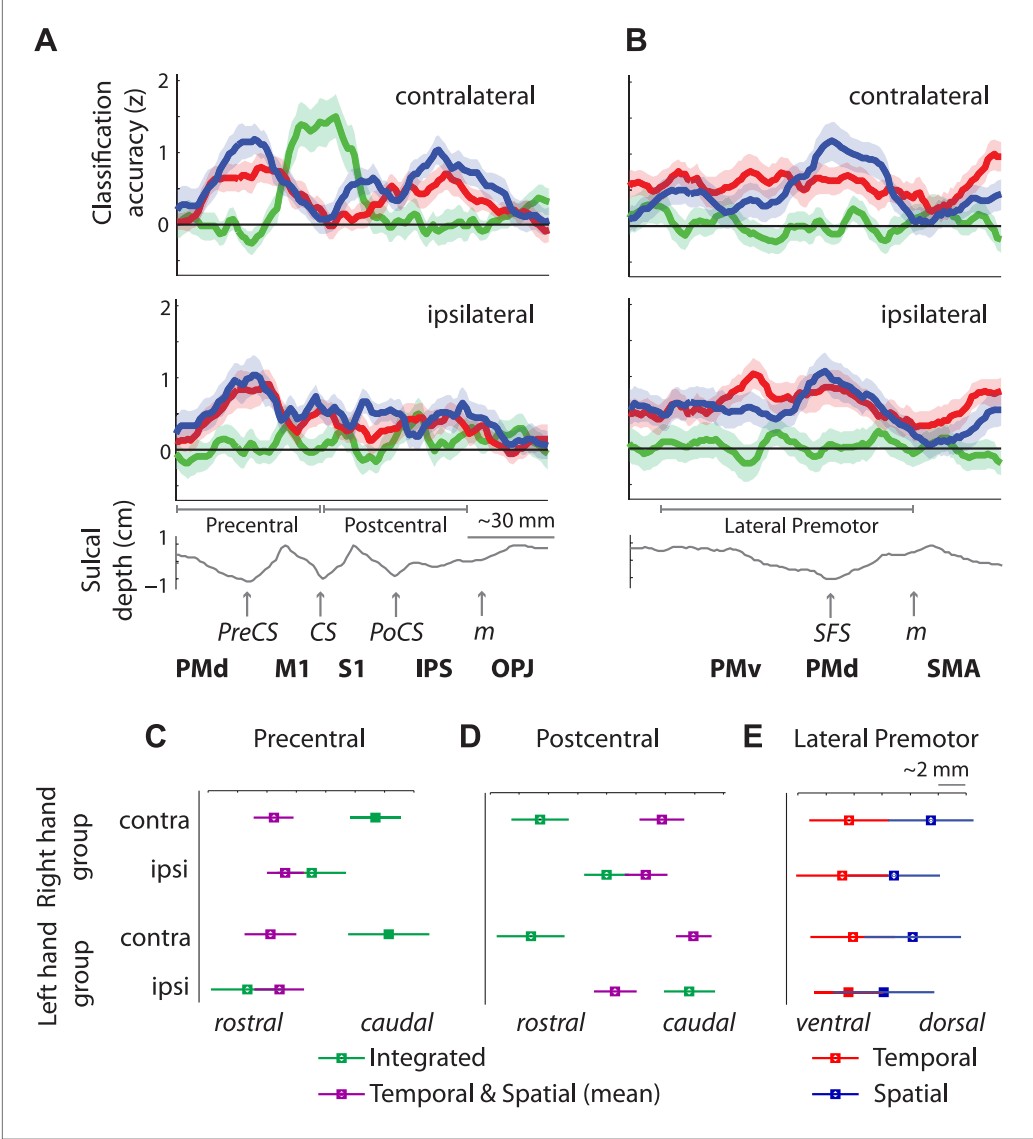

**Figure 6**. Distribution of encoding in cortical cross-sections. Shown are profiles of integrated (green), temporal (red), and spatial (blue) classification accuracy (z-values), (**A**) on a cross-section running from rostral premotor cortex, through the hand area, to the occipito-parietal junction and (**B**) on a profile running from the ventral, through the dorsal premotor cortex, to the SMA (BA 6). (**C**) Center of gravity (CoG) analysis across the precentral part (between rostral PMd and central sulcus) of the profile in **A** shows that independent temporal and spatial classification accuracy (mean in purple) is represented more rostrally to integrated classification accuracy in the contralateral hemisphere across right and left hand groups. (**D**) CoG analysis across the postcentral part of the profile shows the opposite pattern to **C** with independent classification accuracy represented more caudally, further away from the CS towards the parietal cortex as compared to integrated classification accuracy. This gradient was found in the contralateral hemisphere across right and left hand groups. (**E**) CoG analysis across the lateral premotor cortex shows a slight ventral bias for temporal compared to spatial classification accuracy across hemispheres and groups. BA, Brodmann area; CoG, center of gravity; IPS, inferior parietal sulcus; m, medial wall; M1, primary motor cortex; OPJ, occipito-parietal junction; PMd, dorsal premotor cortex; PMv, ventral premotor cortex; PoCS, postcentral sulcus; PreCS, precentral sulcus ventral premotor cortex; S1, primary sensory cortex; SFS, superior frontal sulcus; SMA, supplementary motor area.

The following figure supplement is available for figure 6:

**Figure supplement 1**. Distribution of encoding as in *Figure 6* split by group (**A**) on a cross-section running from rostral premotor cortex, through the hand area, to the occipito-parietal junction and (**B**) on a cross-section running from the ventral, through the dorsal premotor cortex, to the SMA (BA 6).

different temporal profiles. However, the response derivate also allowed us to test for the information contained in different temporal activation profiles (*Figure 4—figure supplement 3B*). As expected, based on the derivative, we observed increased classification accuracy of independent spatial and to some degree also temporal features in M1, while in premotor areas such as the PMd, the independent classifiers performed much worse, in line with the evidence suggesting weaker finger representations in premotor areas (*Diedrichsen et al., 2013b*). The factors response type (main response vs derivative), ROI (M1 vs PMd), and hemisphere (contralateral vs ipsilateral) showed a significant interaction between the three factors for both spatial ($F_{(1,31)}$ = 9.165, p=0.005) and temporal encoding ($F_{(1,31)}$ = 5.395, p=0.027). This suggests that the voxel patterns based on the main response estimates are unlikely to reflect differences in the temporal profile of the observed response.

Finally, the above chance classification could reflect simple differences of movement parameters during the sequence execution rather than sequence encoding (*Todd et al., 2013*). Despite employing the same fingers and the same temporal intervals across all nine sequences, as well as by controlling the number of runs and jumps between finger digits and intervals (ascending or descending interval transitions), in some subjects minor, but systematic finger force differences between the trained sequences occurred, such as more force on the thumb in one sequence and on the index finger in a different sequence (*Figure 4—figure supplement 4*). Accordingly, force on the five fingers could be used to reliably classify the nine-trained sequences (mean *zacc* = 2.25, *t*(31) = 10.914, p<0.001). Importantly, however, the strength of force differences did not correlate with classification accuracy in contralateral M1 (*r* = −0.210, p=0.257), such that simple differences in finger forces could not account for the finding of integrated feature encoding here.

Instead, we hypothesized that the reported multivariate encoding of sequences in contralateral M1 would covary with the degree with which that participant showed sequence-specific learning, defined as the RT advantages for trained as opposed to untrained sequences at post-test. Indeed, the classification accuracy correlated with the amount of sequence-specific learning, (*r* = 0.468, p=0.008). Thus, participants with higher behavioural learning effects also showed higher classification accuracy (*Figure 7A*). No positive relationship could be revealed for ipsilateral M1 and either force differences or sequence learning (*r* <−0.222, p>0.186, *Figure 7B* for correlation with sequence learning). This further supports that encoding in contralateral M1 is likely to be related to the sequential skill level.

## Discussion

Our study employed fMRI multivoxel pattern analysis that reflects the differential tuning of individual voxels (*Kamitani and Tong, 2005*; *Kriegeskorte et al., 2006*) to identify neural representations of spatial and temporal finger sequence features. We were able to dissociate independent feature

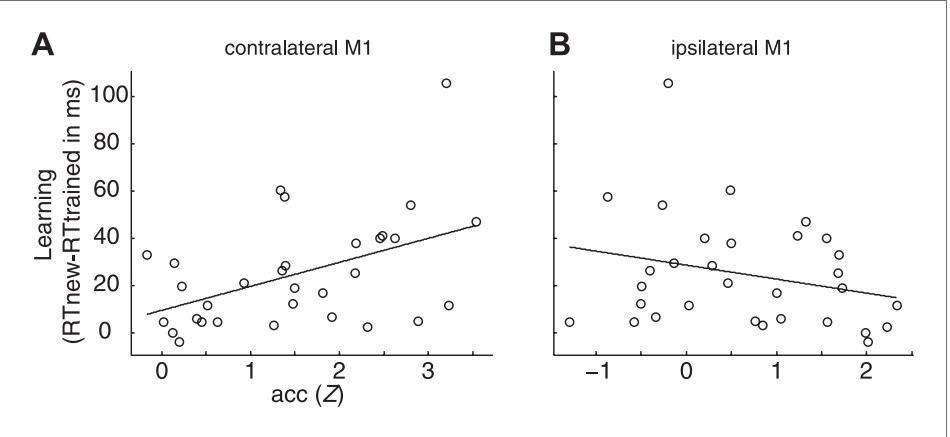

**Figure 7**. Correlation between sequence-specific learning (RT advantages for trained relative to untrained sequences in the post-test) and overall encoding in M1. Learning significantly covaried with the overall encoding in the contralateral M1 *r* = 0.47, p=0.008 (**A**), but not in the ipsilateral M1 r = −0.25, p=0.169 (**B**). The correlation of sequence learning and contralateral encoding in M1 remained significant when taking all, and not only the task-activated voxels in contralateral M1 into account, (r = 0.369, p=0.041).

representations in which voxel patterns related to spatial and temporal sequence features combined linearly, from integrated feature representations in which each spatio-temporal combination was associated with a unique activity pattern. We demonstrate that only the output stage of the cortical motor hierarchy, the primary motor cortex (M1) contralateral to the moving hand, encoded spatio-temporal features of finger sequences in an integrated fashion. In contrast, bilateral medial and lateral premotor cortices showed partly overlapping, but mutually independent representations of the spatial and temporal features. The independent encoding of sequence features in higher order motor areas paralleled our behavioural findings—the nervous system's ability to flexibly transfer both spatial and temporal features from trained to new sequence contexts.

The integrated sequence encoding found in the contralateral M1 is in line with electrophysiological data showing that 40% of neurons in the primary motor area in monkeys can exhibit tuning to sequences of muscle commands (*Matsuzaka et al., 2007*), evidence that inactivation of M1 via muscimol can selectively disrupt sequential behaviour (*Lu and Ashe, 2005*), as well as previous sequence learning studies in humans (*Karni et al., 1995*; *Penhune and Doyon, 2005*; *Steele and Penhune, 2010*). We found that the overall sequence encoding in the contralateral M1 covaried with the amount of behavioural advantages for the trained sequences, suggesting that our analysis uncovered skill-dependent representations. The fact that each spatio-temporal sequence combination had its unique activity pattern in M1 is consistent with a dynamical systems view which proposes that each movement is controlled by a subpopulation of neurons that form a dynamical network (*Laje and Buonomano, 2013*; *Shenoy et al., 2013*). Instead of representing movement features separately, these networks are assumed to produce complex movement patterns based on a neural state-space trajectory, which is determined by the internal connectivity and external input to the circuitry (*Shenoy et al., 2013*). Accordingly, for each unique spatio-temporal sequence a slightly different distribution of neurons is activated in M1 which in turn cause distinct voxel activity patterns for each of the studied sequence combinations (*Kamitani and Tong, 2005*; *Kriegeskorte et al., 2006*). This integrated encoding in M1 is in line with our model, which suggests that the temporal and spatial sequence features are integrated non-linearly in the nervous system (*Kornysheva et al., 2013*).

While adequate for learning and producing specific spatio-temporal sequences, integrated encoding such as found in M1 alone would not allow the system to use learned spatial or temporal features independently. However, subjects showed behavioural advantages for untrained sequence in which only one of the trained features (spatial or temporal) was retained. With spatial and temporal features of movement trajectories emerging from the same local circuits, an integrated representation as proposed in dynamical systems models (*Shenoy et al., 2013*) cannot explain the flexible transfer or independent adaptation of spatial and temporal features reported here and in previous studies (*Ullén and Bengtsson, 2003*; *Ali et al., 2013*; *Kornysheva et al., 2013*). In contrast, our results indicate that higher order motor areas (lateral and medial premotor cortices) parsed the sequences into the two constituent features, in line with the modularity and flexibility we observed in behaviour. The independent albeit partly overlapping spatial and temporal encoding suggests a modular feature-separating storage for movement production.

Although our experiment provides both behavioural and imaging evidence for the independent representation of temporal and spatial features of movements, it is possible that not all classes of movements lend themselves to such a separation. For example, the temporal profile of a force perturbation during a reaching movement is learned inseparably from the whole spatio-temporal trajectory (*Conditt and Mussa-Ivaldi, 1999*). One critical factor that distinguishes these classes may be whether movement kinematics are continuous as in reaching movements or fall into discrete phases, as induced by the current task (*Ivry et al., 2002*). In contrast, for discrete sequential movements, as studied here, the evidence for an independent representation of the temporal structure is compelling. Importantly, our behavioural finding is unlikely to be an effect artificially induced by the orthogonal design, in which all three spatial sequence features were crossed with three temporal features, since flexibility in independently adapting spatial or temporal features of sequences has also been observed in a previous study which involved the training of only one particular spatio-temporal combination (*Kornysheva et al., 2013*).

Previous data demonstrated that the premotor cortex modulates its activity once movement sequences are made more complex in either their spatial or temporal structure (*Bengtsson et al., 2004*), or in the presence of both features (*Sakai et al., 2002*; *Brown et al., 2013*). However, these studies did not establish how temporal and spatial features were encoded. Here, we address this issue,

starting from a recent study that showed that voxel pattern analysis can uncover sequence representations in motor and premotor areas, which are specifically enhanced for trained sequences (*Wiestler and Diedrichsen, 2013*). This classification analysis was now extended to identify regions with voxel patterns related to spatial and temporal sequence features that were represented independently from the respective other feature. Our representational analysis allowed us for the first time to distinguish between integrated vs independent representations, even if the independent spatial and temporal representations overlapped spatially. While we validated this approach by voxel pattern simulations, the analysis hinges critically on the assumption that activity in independent neuronal populations within a single voxel combine additively in the observed BOLD signal. Is this assumption justified? Functional imaging signals show clear non-linearities when studied over a large dynamic range (*Logothetis et al., 2001*). However, the activity of single voxels only varied very slightly between different sequences in our paradigm. For this restricted range, therefore, we can be relatively confident that a linear approximation is reasonable (*Diedrichsen et al., 2013a*). Furthermore, because any non-linearity between neural signal and hemodynamic response should be relatively homogenous across adjacent cortical motor areas, the distinct representational dissociation found between M1 and PMd is likely to reflect neural rather than hemodynamic differences.

How can the independent encoding of spatial and temporal features be implemented on a single cell level? Neurons in primate premotor cortices have been shown to be tuned to specific spatial transitions between movement elements and whole sequences of movements increasing their firing rate prior to a movement based on its sequential context (*Mushiake et al., 1991*; *Tanji and Shima, 1994*; *Shima and Tanji, 2000*). Such units have been suggested to be an important component in the organisation of skilled sequential behaviours, however whether timing between movements modulates the same neurons has not been systematically explored. Our findings of independent representations of the spatial and temporal features predict that the representation of movement order in space first reported by Shima and Tanji in the SMA (*Tanji and Shima, 1994*) is likely to be independent of the exact timing of individual movements in the sequence. In other words, changing the temporal intervals between movements in a sequence should not interfere with the tuning of the neuron to a specific spatial movement transition or sequence.

At the same time our findings predict independent encoding of the temporal features of movement sequences within the same areas as the spatial features. In the medial premotor cortex of monkeys, *Merchant et al. (2013)* found evidence for the encoding of sub-second intervals sensitive to the sequential context within a synchronization-continuation task. In addition to neurons tuned to interval duration, the authors found cells that are tuned to both a specific interval between movements and a specific position in a sequence, for example, a neuron increased its firing rate only to the 850 ms interval between the fifth and sixth lever push. This type of encoding would be detectable with our method since in the trained sequences the individual digits and temporal intervals occurred at unique positions. However, the intervals tested in each sequence were isochronous, and it is currently unknown whether this type of encoding would generalize to a sequential temporal pattern similar to the one employed in the current task.

Furthermore, the timing of a motor responses can be related to the firing of neural units in the cerebellum, as for instance in the case of cerebellar Purkinje cells which decrease simple spike activity in a timed fashion depending on the trained interval between the conditioning stimulus and the conditioned response following training (*Jirenhed and Hesslow, 2011a*, *2011b*). Yet, whether this type of encoding can also be modulated by the sequential context is not known. In contrast to our findings in the cortex, the cerebellum did not yield significant results with regard to temporal encoding, although there was a clear significant overall encoding of the sequences in the ipsilateral lobule VI which forms reciprocal connections with contralateral premotor areas (*Buckner et al., 2011*; *Bostan et al., 2013*). This null result could be related to higher noise levels for subtentorial structures (*Wiestler et al., 2011*) or the more complex folding structure of the cerebellar cortex, which may push the informative signal variations below the threshold of effective spatial resolution.

Despite partly overlapping temporal and spatial representations, we found a gradient with spatial features represented more in dorsal and temporal features more in ventral aspects of the lateral premotor cortex similar to studies involving spatial and temporal production (*Bengtsson et al., 2004*) and prediction (*Schubotz and von Cramon, 2001*). This regional distribution raises the possibility that the temporal representation uncovered here may not be an abstract representation of the temporal structure of sequences, but rather a representation of an additional effector system. Indeed, most subjects'

introspective reports suggests subvocal rehearsal at the beginning of their training, albeit less so towards the end of training and during fMRI. However, a simple mapping of the temporal feature representations to the control of vocal sequencing and timing is unlikely. There was no direct involvement of more posterior and perisylvian primary vocalization centres. Moreover, cross-sections through the cortical Brodmann area 6 suggest that temporal encoding largely overlapped with spatial encoding along the premotor cortex, instead of being restricted to ventral-most premotor areas recruited for rhythmically structured vocal rehearsal (*Riecker et al., 2002*). Instead, the nervous system may build on specialised processes in premotor areas for temporal control, which have originally evolved for the sequencing of the oro-facial and laryngeal musculature in speech production (*Schubotz, 2007*). Such a temporal representation in the premotor cortex could modulate the finger motor system at the level of M1 by interacting with premotor input from the spatial representation in a non-linear manner, as suggested in our multiplicative model (*Kornysheva et al., 2013*), thereby modulating the integrated dynamical systems representation of the sequences in M1.

The presence of multiple temporal and spatial representations across the cortex suggests parallel computations related to the same sequential features. These may have complimentary functions. The medial vs lateral premotor cortex encoding of sequence features may be related to the concurrent encoding of both internally and externally (visually) driven sequential movements (*Goldberg, 1985*; *Mushiake et al., 1991*). Within the lateral premotor cortex, temporal processing has been hypothesized to be associated with the degree of motor involvement; *Chen et al. (2009)* have shown that the ventral premotor cortex enables direct action-related encoding of temporal structure while the dorsal premotor cortex facilitates higher order temporal organisation of sequences, in line with the direct vs indirect transformation hypothesis by *Hoshi and Tanji (2007)*. Finally, multiple spatial representations may reflect different spatial sequence reference frames—such as movement sequences in extrinsic spatial coordinates in the rostral PMd and the posterior parietal cortex (*Brown et al., 2013*; *Wiestler et al., 2014*) and in intrinsic spatial coordinates in the caudal PMd (*Wiestler et al., 2014*). Taken together, such a diversity of spatial and temporal sequence representations across the network of premotor and parietal areas may enable flexible control of skilled behaviour that can adapt to the situational task requirements.

Overall, the independent encoding of the spatial and temporal features of movement sequences in premotor areas endows the nervous system with the ability for adjustments of individual movement parameters—which would not be possible with a fixed integrated representation such as in M1 alone. For example, this separate encoding may explain why a pianist who learned a particular passage can effortlessly produce the same sequence of finger movements with a novel rhythmic structure—or combine the same rhythm with new variations of the sequence of notes.

Finally, the decomposition into features in the premotor cortex provides a computational solution for representing longer and more complex sequences of actions. If the system utilized integrated encoding alone, such as in the primary motor cortex, it would have to represent all relevant permutations of spatial, temporal and other relevant parameters (e.g., amplitude and direction), which very quickly would lead to a combinatorial explosion. Instead, the premotor cortex may dedicate its resources to representing these features separately, creating a more compact and flexible representation. This special feature may explain why evolution endowed us with premotor areas, rather than simply with a larger primary motor cortex. This architecture has parallels in the ventral visual stream, in which lower visual areas encode specific combinations of simple features at specific spatial locations, whereas higher visual areas represent more complex visual arrangements—such as body parts and scenes—independently of their orientation or location of the stimulus (*Freeman and Simoncelli, 2011*). Thus, the decomposition of movements into features may be the cardinal function of premotor areas, and endow the system both with behavioural flexibility and the capacity to store long, complex sequences of movements.

## Materials and methods

### Participants

32 neurologically healthy volunteers took part in this study (16 female), aged between 19 and 36 (mean: 24.8, SD: 5.6). Half of the subjects were trained and scanned on the right and half on the left hand (males and females balanced across groups). All subjects were right-handed according to the Edinburgh Inventory of Manual Preference (*Oldfield, 1971*) with a mean score of 89 (range: 70–100; SD: 11.9; both groups had the same mean score of 89, right group SD: 10.4, left group: SD 13.5). None of them

were professional musicians or athletes. All subjects were naive concerning the hypothesis of this study. Experimental procedures were approved by the research ethics committee of University College London. Written informed consent was obtained from each participant for data analysis and publication of the study results.

## Apparatus

Participants placed all five fingers of either the left or right hand on a keyboard, which was secured with a foam pillow on the participant's lap. The keyboard had five elongated keys, 20 mm wide, with a groove for each fingertip. A force transducer was mounted below each key and measured the force exerted by the fingers. The force transducers (Honeywell FS series) had a dynamic range up to 16 Newton ($N$), with a repeatability of constant force measurements of <0.02 $N$. Signals from the force transducers were transmitted from the scanner room via a shielded cable. Filters in the scanner room wall prevented leakage of radiofrequency noise. In 21 out of 32 subjects, the force of the fingers of the untrained hand was recorded on a keyboard for the other hand to monitor potential mirror movements (*Diedrichsen et al., 2013b*). In the remaining subjects, the passive hand was placed on the pillow on the left or right leg, respectively. Force traces revealed no mirror movements on the contralateral hand.

## Procedure and behavioural task

Participants viewed a projection screen mounted behind the scanner bore via a mirror. The screen showed a central cross, on which participants were instructed to fixate during the entire experiment. Participants executed isometric right or left finger presses against the non-movable keys. We implemented a visually cued motor learning task (*Figure 1*), (*Kornysheva et al., 2013*) to force the subjects into a specific spatial and temporal structure of movement. Subjects were presented with a sequence of white digits (1–5) in the middle of a black screen that was repeated three times in the training and fMRI sessions. The digits 1, 2, 3, 4, and 5 instructed to press the thumb, index, middle, ring, and little finger of the right or left hand, respectively. We expected that while the first execution of a sequence would be rather reactive and driven by the cue, the second and third execution would rely on a learned representation of the sequence in question. Indeed the second and third execution during fMRI was significantly faster than the first one ($t$(31) = 9.608, p<0.001). Participants were instructed to perform the task as fast and accurately as possible. Each trial started with a warning cue ('!'; duration: 400 ms), followed by a sequence of five digits that was timed according to a sequence of five possible inter-stimulus-interval values (ISI; 600, 800, 1000, 1400, 1700 ms). The first ISI commenced when the warning cue disappeared. Each digit remained on the screen until the onset of the next digit according to the respective ISI value or for 600ms after the onset of the last digit. The trial ended with an inter-trial interval of 1.6 s, making each trial 8.1 s long. Subjects received feedback on their performance throughout the experiment as follows: if the subjects pressed the correct button within the limits of 50 ms before the onset of the current and 50 ms before the onset of the next digit, that digit turned green; if the response was too early the next digit appeared in yellow; if the response was too late the digit turned turquoise; if the finger press was incorrect, the digit turned red. Subjects received a point only when all digits in a sequence turned green, that is, when they pressed the correct finger in the correct time window. After each block of 27 (training and post-test) and 54 (fMRI test) trials, respectively, subjects received feedback on their cumulative point score and the median RT in the last block. They were informed that the participant with the highest cumulative score (weighted by their reaction time) would receive an additional financial reward.

Movements were instructed by sequences with a particular combination of digit order and timing (ISI sequence). Ascending or descending digit run triplets (e.g., 2-3-4) were excluded from the pool of possible sequences. Identical triplets across sequences were prohibited. The number of ascending to descending and descending to ascending direction changes was fixed at 2 across all sequences (e.g., in the sequence and 5-2-4-3-1 the direction change is at 2-4 and 4-3). The position of the five elements (finger digits and temporal intervals) had to be different for the three-trained spatial sequences and the three-trained temporal sequences, respectively. The sequences were randomly generated for each participant according to these criteria and matched across the right and the left hand training groups.

The experiment was conducted over 5 consecutive days, with a training (days 1–3), fMRI (day 4), and post-test phase (day 5). Note that one of 32 subjects could not take part in a post-test. 3 of the 32 subjects had a delay of 1–4 days between fMRI and post-test. Finally, one subject had the last training session scheduled on the same day as the fMRI.

The training phase took place on 3 days and took approximately 1.5 hr per day, involving 21 blocks of 27 trials each. 18 of 21 blocks contained the nine different combined sequences (3 spatial × 3 temporal structures) presented three times in a row (mini-blocks). Three additional probe blocks per day were introduced to measure RT advantages related to trained sequences, as well as independent temporal and spatial transfer to new sequences throughout the training phase. Each probe block contained three probe conditions. In the *trained temporal* condition the cues appeared with each of the trained temporal features, but indicating an untrained order of finger presses. The untrained order was different in each of the three timing probe blocks, but was repeated across the three trials of each mini-block. In the *trained spatial* condition, the visual stimulus cued sequences involving trained spatial sequence features of finger presses, but in combination with a new temporal feature, that is an untrained sequence of inter-stimulus-intervals. Equally, the new temporal sequence was different in each of the three order probe blocks, but did not change across the three trials of each mini-block. Finally, the *untrained* condition cued a sequence of finger presses and inter-stimulus intervals that were different from any other trained condition. Again, the novel combination of spatial and temporal features was different in each of the three untrained sequence blocks, but repeated three times in each mini-block. This made the probe blocks as similar as possible to the training blocks. Finally, the three training sessions were identical in terms of sequences and trial randomisation, ensuring that behavioural change could not be explained by differences in sequences or trial delivery. The probe blocks appeared in the beginning (1st block), the middle (11th block) and the end (21st block) of each training session. Whether subjects started the first training day with a probe block or a training block was counterbalanced across subjects. Note that *Figure 2B* displays the first probe block as 1st block and the first training block on 2nd block for all participants.

## Behavioural analysis

Reaction times (RTs) for each response were defined as the time at which the force of a finger reached maximum velocity around the onset of the visual cue. Only correct responses were considered. Also, responses that occurred more than 100 ms before stimulus onset or more than 600 ms after stimulus onset were considered as errors and excluded from further analysis. Within each correct trial, we averaged the RT for all responses. We then used the median RT across trials for each individual and condition in the group analysis. Since one of the subjects in the right hand training group did not participate in the post-test, in the post-test analysis only, we excluded a subject trained on corresponding sequences on the left hand to ensure that sequences employed were matched across groups. However, the results of the post-test analysis did not change qualitatively when these subjects were included. Error rates were determined for each block and condition (cf. *Figure 2—figure supplement 1*).

## Scan acquisition

Data were acquired on a 3 T Siemens Trio system with a 32-channel head coil. Functional data comprised 6 runs of 190 vol each, using a 2D echo-planar imaging sequence (repetition time [TR] = 2.72 s). The first 3 vol were discarded to allow magnetization to reach equilibrium. We acquired 32 slices in an interleaved sequence at a thickness of 2.7 mm (0.3 mm gap) and an in-plane resolution of 2.3 × 2.3 mm$^2$. The matrix size was 96 × 96. Trials were triggered every 2.97 TR (every 95 slices). The slices were positioned to cover the cortical motor and premotor areas, as well as the cerebellum. The ventral prefrontal cortex, anterior temporal lobe, and the superior-most part of the parietal lobe were not covered in each subject. Field maps were obtained after the first functional run to correct for inhomogeneities in the main magnetic field (*Hutton et al., 2002*). We also acquired a single T1-weighted anatomical scan (3D magnetization-prepared rapid gradient echo sequence, 1 mm isotropic, 240 × 256 × 176 mm field of view).

## First-level analysis

The functional data were analysed using SPM8 (http://www.fil.ion.ucl.ac.uk/spm/), and custom written MATLAB code (The MathWorks, Inc., Natick, MA). First, we corrected for slice acquisition timing by shifting the acquisition to align with the middle slice of each volume. We then corrected for head movements using a 6-parameter motion correction algorithm. This step also included correction of possible image distortions using the acquired fieldmap data (*Andersson et al., 2001*; *Hutton et al., 2002*). The realigned functional data were then coregistered to the individual anatomical scan, using the automatic algorithm in SPM. The coregistration was visually checked, and the affine parameters were adjusted by hand to improve the alignment, if necessary.

The preprocessed data were analysed using a general linear model. To remove the influence of movement-related artifacts, we used a weighted least-squares approach (*Diedrichsen and Shadmehr, 2005*). For each of the nine trial types, we defined one regressor per imaging run that captured the activation for each voxel across the three sequence executions. This main regressor consisted of boxcar functions for each execution of the sequence of that type, each starting at the moment of warning cue and lasting for 6.5 s (2.4 TRs), which was then convolved with the standard hemodynamic response function (*Figure 4—figure supplement 3*). Additionally, we included the temporal derivate of the response in the model, linearly independent of the main response. Only the main response was used for classification analysis, to determine differences in spatial activation patterns averaged over the sequence executions, rather than differences in the temporal profiles of the sequence. In addition, each error (incorrect finger response) was modeled as a regressor of no interest, one regressor per imaging run starting at the onset of the digit presentation associated with an erroneous response and lasting for 2.72 s (1 TR) to account for errors at different positions in the sequence. For each imaging run and voxel, the analysis therefore estimated in 20 regression coefficients (9 sequence regressors, 1 error regressor plus 10 respective temporal derivative), from which the nine main sequence response estimates were used in the classification analysis.

## Searchlight approach

The searchlight analysis was performed taking into account each subject's individual anatomy. This approach was used to consider anatomical variations and achieve maximum accuracy in the localization of representation. The cortical searchlight analysis was implemented on each subjects' individual surfaces. The cerebellar searchlight was volume-based within the cerebellar grey matter as defined by SUIT.

From the anatomical images, we obtained a surface reconstruction using the software Freesurfer (*Dale et al., 1999*), which estimates the outer boundary of the gray matter (pial surface) and the white–gray matter boundary (white surface). The surfaces were aligned via spherical registration to the Freesurfer average atlas (*Fischl et al., 1999*). Individual data were then projected onto the group map via the individual surface. Correction for multiple tests was performed on the surface using Gaussian field theory (*Worsley et al., 1996*).

To detect sequence-specific representations in the neocortex, we used a surface-based searchlight approach (*Oosterhof et al., 2011*). The corresponding toolbox is available on http://surfing.sourceforge.net. A circular region was defined on the cortical surface and the radius increased until exactly 160 voxel lay between the selected surface patches on the pial and white surfaces. It has been shown that a surface-based searchlight minimizes the spillover of multivoxel information from one region to the next across a sulcus and therefore allows for more regionally specific inferences (*Oosterhof et al., 2011*) than volume-based searchlights (*Kriegeskorte et al., 2006*). The classification accuracy for each searchlight (cf. classification procedures below) was assigned to the center of each searchlight. A classification accuracy map was generated by moving the searchlight across the cortical surface.

For the identification of sequence-specific representations in the cerebellar cortex, a volume-based searchlight approach was utilized (*Kriegeskorte et al., 2006*; *Wiestler et al., 2011*). Each searchlight consisted of 160 adjacent voxels and the calculations were restricted to voxels lying in the cerebellum using a masking algorithm in the SUIT toolbox (*Diedrichsen, 2006*).

## Multivoxel pattern classifiers

We implemented different classification procedures to extract information regarding the overall, temporal, spatial, and integrated encoding of sequences. We first employed an overall multi-class classifier, which tested for any differences in the activity patterns associated with nine unique classes (combinations of three spatial and three temporal structures) of sequences (*Figure 3A*). The mean pattern for each class, and the common voxel-by-voxel co-variance matrix was determined from the training data set, consisting of five of the six imaging runs. A Gaussian-linear multi-class classifier (for details see *Wiestler et al., 2011* and *Source code 1*) was then used to independently classify the nine patterns of the test data set (the remaining imaging run). By rotating which runs served as training and test set, we obtained a cross-validated classification accuracy. Values above the chance level of 11% (1/9 classes) indicated that there were some systematic differences between the activity patterns.

A region that encodes the order of finger presses (spatial sequence) independently of the temporal sequence should show similar activation patterns for each spatial sequence across the three temporal

sequences (*Figure 3B*). To test for such a representation, we used a linear classifier that distinguished between three spatial sequences paired with one particular temporal sequence, while being trained on data from trials in which spatial sequences were paired with the two remaining temporal sequences. To guarantee independence of the beta estimates, training and test sets were drawn from separate imaging runs. The procedure therefore involved training to distinguish between three spatial sequences timed according to two temporal sequences from five runs and testing the difference between the spatial sequence patterns paired with the remaining different temporal sequence from the remaining run, resulting in a 18-fold cross-validation procedure. Significant above-chance classification accuracy performance (33.3%, 1/3 classes) indicated the presence of systematically different local activation patterns for different spatial sequences. The temporal classifier (*Figure 3C*) was designed following the same principle, simply exchanging the role of spatial and temporal sequence.

Finally, the integrated classifier isolated representations that code for the unique combination of temporal and spatial sequences. Like the overall classifier, it treated each of the nine unique combinations as a separate class. Critically, however, the mean temporal and the mean spatial class patterns of each run were subtracted from the overall pattern of the respective run. Since this was done for every run separately, subtracting the respective activity patterns did not induce dependence between training and test sets (see also *Figure 3E* for simulation results). The residual patterns therefore reflected the interaction component between timing and order that cannot be attributed to a linear combination of the two factors.

For better comparability across classifiers, as well as for group analysis, the classification accuracies were transformed to z-scores, assuming a binomial distribution of the number of correct guesses. We then tested these z-scores against zero (chance level) across participants. Whole brain results were corrected using a surface-based random effects analysis (N = 32) with an uncorrected threshold of $t(31) > 3.37$, $p<0.001$ and a cluster-wise p-value for the cluster of that size. The p-value was corrected over the cortical surface using the area of the cluster (*Worsley et al., 1996*) and further Bonferroni corrected for tests over two hemispheres ($p_{corrected} = p*2$). Significance in the cerebellum was assessed using a small volume correction (SUIT).

## Pattern distance simulations

We validated this classification approach using simulations of activity patterns of 160 voxels. The activity pattern (y) for the $i^{th}$ spatial and $j^{th}$ temporal sequence for the $k^{th}$ imaging run was generated as:

$$\mathbf{y}_{i,j,k} = s_i + t_j + i_{i,j} + r_k + e_{i,j,k}$$

Each pattern component (*Diedrichsen et al., 2011*) was generated as a normal random vector over 160 voxels. By varying the variance of the spatial (s), temporal (t), and integrated (i) representations relative to the noise (e), we obtained simulated data sets that were then submitted to the same classification approach used for the actual imaging data (*Figure 3E*).

## Regions of interest (ROIs)

To examine the distribution of different types of finger sequence representations (spatial, temporal, integrated) across motor areas and a possible interaction with the hand being trained, anatomical regions of interests were defined symmetrically in both hemispheres based on probabilistic cytoarchitectonic maps aligned to the average Freesurfer surface (*Fischl et al., 2008*) as areas with at least 35% probability of the respective field. Four motor areas with significant overall sequence encoding were considered for ROI analysis: bilateral primary motor (M1), supplementary motor area (SMA), dorsal (PMd), and ventral premotor cortex (PMv). The hand region of primary motor cortex (M1) was defined as Brodmann area (BA) 4, 2.5 cm above and below the hand knob (*Yousry, 1997*). Dorsal premotor cortex (PMd) was defined as the lateral aspect of BA 6, superior and PMv inferior to the middle frontal gyrus. The supplementary motor areas (SMA/pre-SMA) comprised the medial aspect of BA6. The M1, SMA, and PMd were identical to the ROIs used in previous work (*Wiestler and Diedrichsen, 2013*) and the PMv was added as the lateral part of the premotor cortex (BA6) ventral to PMd.

We averaged data across all voxels in the anatomically defined ROIs. This approach enabled us to uncover the respective sequence representations in regions independent of their metabolic demand during the task.

A z-score for the classification accuracy for overall, spatial, temporal, and integrated encoding across each in the respective ROI was determined for each subject. ROI analysis was performed on non-smoothed data of each individual. One-sample *t* tests were employed to probe encoding above

chance level (zero). The p-value was Bonferroni-corrected for eight comparisons in each ROI (critical p-value was 0.006=0.05/8).

## Cross-sections and center of gravity (CoG) analyses

In addition to the ROI approach, we defined two symmetrical cross-sections to investigate the continuous profile of sequence feature encoding across cortical regions of both the contra- and ipsilateral hemispheres. The first cross-section through the surface map ran from the rostral end of PMd to the posterior superior parietal cortex. The second cross-section went through BA6, from the ventral tip through dorsal premotor cortex into the SMA. To determine biases in sequence feature encoding within these cross-sections, we determined the center of gravity (CoG) for each subject, classification type, and hemisphere. The minimum classification accuracy value of the respective classifier across the profile of interest was first set to zero (baseline) before determining the CoG of the accuracy shapes. The CoG was calculated by computing the spatial average of the coordinates of all nodes in the cross-section, after weighting each with the normalised classification accuracy z at this point. The CoG analysis was performed for three subsections: (1) rostral PMd to central sulcus, (2) central sulcus to the lateral part of the superior parietal cortex hitting the medial wall, and (3) The lateral aspect of BA6, starting at the level of the inferior frontal gyrus up to the crown of the cortex, excluding the portion of BA6 in the medial wall.

## Multivoxel encoding and behaviour

We set out to probe the relationship between the amount of sequence learning across subjects and the encoding of the sequences in the contralateral primary motor cortex (M1) that our classification approach revealed to be involved in integrating the spatial and temporal features of sequences, as well as the ipsilateral M1 as a control region. We considered the 20% most activated voxels in contralateral and ipsilateral M1, thus taking only nodes that were most recruited during the task. Sequence-specific learning was defined as the individual RT advantage for the trained sequences compared to untrained sequences in the post-test phase (cf. RT results). However, we also considered the possibility that the classification results may have been influenced by systematic differences in the execution of the sequences, specifically by differences in force produced by each finger. A region that is sensitive to these lower-level behavioural differences may look like it encodes sequential aspects of the task. To quantify the differences between sequences along these behavioural variables, we computed the accuracy of multivariate classification of the nine trained sequences (overall classifier), using maximum forces at each of the five fingers as different data features instead of voxel activity. Higher classification accuracy would indicate more pronounced force differences between the sequences. Subsequently, two-sided Pearson's correlations between the overall encoding in the contralateral and ipsilateral M1 and the RT difference between untrained and trained sequences in the post-test (RT advantage), as well as force classification accuracy were calculated.

## Acknowledgements

We thank Neil Burgess for constructive comments on the multivariate classification analysis, Tobias Wiestler, Alexandra Reichenbach, and Jamaine Stacey for helping with fMRI data acquisition, Bence Olveczky, Atsushi Yokoi, and George Prichard for helpful comments on the manuscript. This study was supported by the Marie Curie Initial Training Network 'Cerebellar–cortical control: cells, circuits, computation, and clinic' (JD), a Sir Henry Wellcome Fellowship (KK; 098881/Z/12/Z), and strategic award to the Wellcome Trust Centre for Neuroimaging (091593/Z/10/Z), both from the Wellcome Trust.

## Additional information

### Funding

| Funder | Grant reference number | Author |
|---|---|---|
| Wellcome Trust | 098881/Z/12/Z | Katja Kornysheva |
| European Commission | Marie Curie Initial Training Network | Jörn Diedrichsen |

The funders had no role in study design, data collection and interpretation, or the decision to submit the work for publication.

## Author contributions

KK, Conception and design, Acquisition of data, Analysis and interpretation of data, Drafting or revising the article, Contributed unpublished essential data or reagents; JD, Conception and design, Analysis and interpretation of data, Drafting or revising the article, Contributed unpublished essential data or reagents

## Ethics

Human subjects: Experimental procedures were approved by the research ethics committee of University College London. Written informed consent was obtained from each participant for data analysis and publication of the study results.

## Additional files

### Supplementary file

• Source code 1. Custom written multiclass classification function in Matlab which uses linear discriminant analysis. Based on a training data set (such as beta values of runs 1–5 in a region of voxels) where the class identity is known, this classification function predicts the respective class identity of each datapoint in a test set (beta values of run 6 in the same region of voxels), for example, a unique combination of spatial and temporal sequence features in a sequence. Following training-test cross-validation ('Materials and methods') the accuracy can be determined by computing the percentage of datapoints that were classified correctly.

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
