## [Decision Letter]

Thank you for sending your work entitled “Human premotor areas parse sequences into their spatial and temporal features” for consideration at *eLife.* Your article has been favorably evaluated by Eve Marder (Senior editor) and 3 reviewers, one of whom is a member of our Board of Reviewing Editors.

The following individuals responsible for the peer review of your submission have agreed to reveal their identity: Jody Culham (Reviewing editor); Virginia Penhune (peer reviewer).

The Reviewing editor and the other reviewers discussed their comments before we reached this decision, and the Reviewing editor has assembled the following comments to help you prepare a revised submission.

Both reviewers and the Reviewing editor agreed that this interesting paper addresses a longstanding issue in motor performance using state-of-the-art functional imaging techniques. There was broad enthusiasm for the experimental question, design and approach.

1) The manuscript requires a revision to resolve one outstanding issue. The revision should address the fact that the behavioral data showed negligible differences between trained temporal sequences and untrained sequences. These results should be considered in the context of past fMRI studies (including the literature on rhythm/melody and speech sequences). It is also worth reconsidering whether the claim that temporal sequences can be dissociated from spatial ones is fully justified. Although the manuscript emphasizes M1 and PMC, dissociations between spatial and temporal coding were observed in other areas (PPC and ACC) and warrant further consideration.

2) In addition, the manuscript would benefit from revisions to (a) provide a clearer motivation at the outset for having two groups using the right hand or left hand; (b) clarify whether other subcortical structures (e.g., basal ganglia, hippocampus) were included in the search volume; and (c) provide additional discussion of the behavioral performance deterioration in the fMRI session compared to earlier sessions (e.g., restricted mobility, supine positioning and consequent change in frame of reference, apparatus, time of day).

As one reviewer clearly articulated further details regarding point #1, their specific comments are provided below.

Reviewer #2:

Although I am very enthusiastic about the paper, there are two main issues that I think needs to be addressed. First, the behavioral results do not seem to me to support the strong interpretation of separability of spatial and temporal information put forward by the authors. The same issue is present in the interpretation of the imaging results, where the authors argue for separate encoding in the PMC. In contrast, they neglect what appear to be greater differences between the two conditions in parietal cortex and ACC. I think that they could strengthen their interpretation of the current findings and the integration of these findings with their own theoretical model by considering the regions that encode the two different streams of information in a broader, network context. Finally, I think that they could better integrate their findings with previous human fMRI studies of motor sequence learning as well as those related to melody and rhythm processing.

Results: Behavioral data

I am not entirely convinced about the separable learning of the temporal sequences in this experiment. The results of the learning days show no differences between the blocks with the trained temporal sequence and the untrained spatial/temporal sequence. The final day of testing does show differences, but only after 3 trials. Participants also begin at the same level as the untrained sequences, whereas the spatial sequences show savings. If you dropped the first trial and got rid of the variability in the average, would there be any differences?

Overall, I am not so convinced that these results are so different than those of Shin and Ivry or of Brown et al. who both show that the most significant learning and transfer is for the spatial order (or melody, in the case of Brown) and that the temporal order can only follow learning of the spatial order. I think that this makes sense from the point of view of the authors own theory and from that of others (see Hikosaka and Steele and Penhune). Essentially, you cannot accurately time a movement with an indeterminate target or trajectory. I think that this fits with the author's concept of temporal information being a multiplicative go signal. The examples the authors give of musical performance and speech also fit with this. In the first case, the musician already knows the melody (spatial order) and thus can modify the rhythm. In the case of the speech examples: 1) changing word order has to change timing by definition, and 2) temporal prosody is again a rhythm change laid on top of a known sequence of target movements. Overall, this experiment provides evidence that temporal information can be encoded as a separate stream of information, but whether it can actually be separated from a movement sequence in execution is a different question. I don't think that the authors have to make such a strong separability case to have this be very interesting and useful experiment.

Results: fMRI data

Similar to the behavioral results, the authors emphasize the separability of the different brain regions the encoding of spatial, temporal and combined results. In particular, they emphasize the fact that spatial and temporal features appear to be somewhat differently distributed along the PMC, but they do not talk about what appears to be the bigger difference between the two conditions in the parietal lobe (spatial > temporal) and the ACC (temporal > spatial). The fact that the parietal lobe seems to be exclusively tapped by the spatial condition fits with the well-described role of parietal cortex in spatial to motor transformations. It is also consistent with the results of Brown et al. who showed that intra-parietal sulcus showed greater repetition suppression response to melodies vs. rhythms. The parietal contribution to the spatial component could fit with the authors' theory in that it describes the system or mechanisms by which spatial information is encoded when the sequential and spatial information are not fully integrated. Similarly, the ACC engagement in the temporal condition seems to fit well with the idea that temporal information is some kind of a go signal operating on top of a spatial/kinematic sequence given the ACC's role in motor reprogramming and response inhibition.

Finally, the fact that voxels encoding the two streams of information in PMC and SMA largely overlap suggests that these regions may be engaged in more general processes linking stimulus to response and being the final common pathway for both spatial and temporal information to reach M1. This idea fits more generally with ideas about the role of the PMC in motor control and performance.

---

## [Author Response]

*1) The manuscript requires a revision to resolve one outstanding issue. The revision should address the fact that the behavioral data showed negligible differences between trained temporal sequences and untrained sequences. These results should be considered in the context of past fMRI studies (including the literature on rhythm/melody and speech sequences). It is also worth reconsidering whether the claim that temporal sequences can be dissociated from spatial ones is fully justified*.

Thank you for raising this important point. We agree that there was no difference between trained temporal and untrained sequences in the first three trial repetitions and that the advantage of temporal transfer does only emerge after three repetitions. We have now emphasized this point more clearly in the text. Importantly, this finding is an exact replication of our previous experiments ([31], Figure 3) showing that the relative behavioural difference between trained temporal and untrained sequences emerges with further trial repetitions of both types of sequences. This delayed behavioural advantage suggests the existence of an independent representation of the temporal structure in the CNS. However, this representation can only be utilized when the spatial sequence is known. In contrast, a known spatial sequence leads to immediate advantage in reaction time. In [31] we offer a simple model to explain this asymmetry: Both temporal and spatial sequences are represented independently, but at the output stage, the temporal representations acts as a modulator on a spatial expectation. Thus, without a spatial expectation, temporal knowledge does not have an effect. This model reconciles disparate results in the literature with regard to temporal transfer (for references, cf. [31]).

Although the manuscript emphasizes M1 and PMC, dissociations between spatial and temporal coding were observed in other areas (PPC and ACC) and warrant further consideration.

We now consider the spatial and temporal feature encoding in these areas in the Discussion.

*2) In addition, the manuscript would benefit from revisions to (a) provide a clearer motivation at the outset for having two groups using the right hand or left hand; (b) clarify whether other subcortical structures (e.g., basal ganglia, hippocampus) were included in the search volume; and (c) provide additional discussion of the behavioral performance deterioration in the fMRI session compared to earlier sessions (e.g., restricted mobility, supine positioning and consequent change in frame of reference, apparatus, time of day)*.

In the revised manuscript, (a) we now outline that having two groups was important to probe whether possible differences between hemispheres reflected hemispheric specialisation or the difference between contra versus ipsilateral encoding. (b) The parahippocampal area was included in the reconstruction of the cortical surface, but did not show any significant encoding. The hippocampus and the basal ganglia were not included in the searchlight analysis since this has been performed on the cortical surface and the cerebellar volume only. We also conducted a volume-based search light over the whole brain; however, encoding accuracies in these subcortical areas were relatively low and did not allow for a dissociation of spatial vs. temporal encoding. (c) In the revised version of the manuscript we also provide possible explanations for the slowing of reaction time in the fMRI session, however, please note that the subjects were trained in a supine position in a mock scanner setup during the training sessions and the post-test.

*As one reviewer clearly articulated further details regarding point #1, their specific comments are provided below*.

*Reviewer #2*:

*Although I am very enthusiastic about the paper, there are two main issues that I think needs to be addressed. First, the behavioral results do not seem to me to support the strong interpretation of separability of spatial and temporal information put forward by the authors. The same issue is present in the interpretation of the imaging results, where the authors argue for separate encoding in the PMC. In contrast, they neglect what appear to be greater differences between the two conditions in parietal cortex and ACC. I think that they could strengthen their interpretation of the current findings and the integration of these findings with their own theoretical model by considering the regions that encode the two different streams of information in a broader, network context. Finally, I think that they could better integrate their findings with previous human fMRI studies of motor sequence learning as well as those related to melody and rhythm processing*.

*Results*: *Behavioral data*

*I am not entirely convinced about the separable learning of the temporal sequences in this experiment. The results of the learning days show no differences between the blocks with the trained temporal sequence and the untrained spatial/temporal sequence*. *The final day of testing does show differences, but only after 3 trials. Participants also begin at the same level as the untrained sequences, whereas the spatial sequences show savings. If you dropped the first trial and got rid of the variability in the average, would there be any differences?*

*Overall, I am not so convinced that these results are so different than those of Shin and Ivry or of Brown et al. who both show that the most significant learning and transfer is for the spatial order (or melody, in the case of Brown) and that the temporal order can only follow learning of the spatial order. I think that this makes sense from the point of view of the authors own theory and from that of others (see Hikosaka and Steele and Penhune). Essentially, you cannot accurately time a movement with an indeterminate target or trajectory. I think that this fits with the author's concept of temporal information being a multiplicative go signal. The examples the authors give of musical performance and speech also fit with this. In the first case, the musician already knows the melody (spatial order) and thus can modify the rhythm. In the case of the speech examples: 1) changing word order has to change timing by definition, and 2) temporal prosody is again a rhythm change laid on top of a known sequence of target movements. Overall, this experiment provides evidence that temporal information can be encoded as a separate stream of information, but whether it can actually be separated from a movement sequence in execution is a different question. I don't think that the authors have to make such a strong separability case to have this be very interesting and useful experiment*.

The reviewer’s comment highlighted that we may not have sufficiently explained our behavioural findings and relied too strongly on our previous publication describing and interpreting this effect (31). In the revised manuscript we extended and clarified why we think that spatio-temporal sequence learning can actually lead to separate representations of the spatial and temporal features. Indeed, the first trials (1-3) of the temporal condition, when the spatial sequence is new, are consistent with the findings by Shin and Ivry, and [6]. These studies probed the effect of temporal learning when the spatial sequence switched on every, or every other, sequence production trial or when paired with random spatial sequences. We could replicate this effect in a control condition, in which the trained temporal sequence was paired with new spatial sequences that changed with every trial repetition – here there were no saving related to temporal sequence training, neither in the early, nor in the later repetition phase (Figure 3E in [31]).

On its own these findings suggest that there is no separate temporal representation that is independent of the spatial representation. However, in the same study we also observed (Figure 3B and E in [31]) that performance advantages of the same trained temporal structure emerge once the novel spatial structure is repeated instead of changed randomly from trial to trial (relative to baseline which is also repeated). In other words we found in three different experiments that not immediately but with repetition having trained the temporal features of a sequence does provide advantages as opposed to a completely new spatio-temporal sequence.

From these behavioural findings we can draw two conclusions: 1) The CNS does form a separate representation of the temporal features of sequences. 2) This representation is integrated with the spatial representation in a multiplicative way. This fits very well with the reviewer’s comment that emphasized that unless an indeterminate target or trajectory becomes apparent, the temporal advantages will remain dormant, which we incorporated in the revised manuscript.

*Results*: *fMRI data*

*Similar to the behavioral results, the authors emphasize the separability of the different brain regions the encoding of spatial, temporal and combined results. In particular, they emphasize the fact that spatial and temporal features appear to be somewhat differently distributed along the PMC, but they do not talk about what appears to be the bigger difference between the two conditions in the parietal lobe (spatial > temporal) and the ACC (temporal > spatial). The fact that the parietal lobe seems to be exclusively tapped by the spatial condition fits with the well-described role of parietal cortex in spatial to motor transformations. It is also consistent with the results of Brown et al. who showed that intra-parietal sulcus showed greater repetition suppression response to melodies vs. rhythms. The parietal contribution to the spatial component could fit with the authors' theory in that it describes the system or mechanisms by which spatial information is encoded when the sequential and spatial information are not fully integrated. Similarly, the ACC engagement in the temporal condition seems to fit well with the idea that temporal information is some kind of a go signal operating on top of a spatial/kinematic sequence given the ACC's role in motor reprogramming and response inhibition*.

*Finally, the fact that voxels encoding the two streams of information in PMC and SMA largely overlap suggests that these regions may be engaged in more general processes linking stimulus to response and being the final common pathway for both spatial and temporal information to reach M1. This idea fits more generally with ideas about the role of the PMC in motor control and performance*.

Thank you for this this interesting suggestion, which stimulated our interpretation in the Discussion. We also further emphasized that our claim of independence of spatial and temporal representations does not rely necessarily on the finding that they are represented in separate areas. Indeed, the advantage of our analysis approach is that we can identify integrated vs. separate representation by virtue of the fact that independent representations the temporal and spatial patterns combine linearly (as found in PMC), and that for integrated representations unique activity patterns are found for each combinations (i.e. temporal and spatial patterns combined non-linearly). Thus, even though spatial and temporal representations were regionally overlapping, including those in the ACC/SMA and the parietal lobe, as best seen on the cross-section through the cortex, our analysis still provides strong evidence that they were represented within overlapping regions in an independent fashion.